# Various-Order Low-Pass Filter with the Electronic Change of Its Approximation

**DOI:** 10.3390/s23198057

**Published:** 2023-09-24

**Authors:** Lukas Langhammer, Roman Sotner, Radek Theumer

**Affiliations:** 1Department of Electrical Engineering, Faculty of Military Technology, University of Defence, 60200 Brno, Czech Republic; 2Department of Telecommunications, Faculty of Electrical Engineering and Communication, Brno University of Technology, 61600 Brno, Czech Republic; 3Department of Radio Electronics, Faculty of Electrical Engineering and Communication, Brno University of Technology, 61600 Brno, Czech Republic; sotner@vut.cz (R.S.); 223283@vut.cz (R.T.)

**Keywords:** Butterworth approximation, Bessel approximation, elliptic approximation, Chebyshev approximation, current-mode, electronic adjustment, frequency filter, reconnection-less reconfiguration

## Abstract

The design of a low-pass-frequency filter with the electronic change of the approximation characteristics of resulting responses is presented. The filter also offers the reconnection-less reconfiguration of the order (1st-, 2nd-, 3rd- and 4th-order functions are available). Furthermore, the filter offers the electronic control of the cut-off frequency of the output response. The feature of the electronic change in the approximation characteristics is investigated for the Butterworth, Bessel, Elliptic, Chebyshev and Inverse Chebyshev approximations. The design is verified by PSpice simulations and experimental measurements. The results are also supported by the transient domain response (response to the square waveform), comparison of the group delay, sensitivity analysis and implementation feasibility based on given approximation. The benefit of the proposed electronic change in the approximation characteristics feature (in general signal processing or for sensors in particular) is presented and discussed for an exemplary scenario.

## 1. Introduction

Active analog frequency filters represent an important part in sensor applications [1] (electrocardiographic systems, phase-sensitive detection, biosensors, for instance). The design procedure of (analog) frequency filters, as one of the fundamental building blocks of many applications in various industry branches, deals with many inherent problems. Frequency filters are implemented to approximate the ideal characteristic (immediate turnover of pass-band area to stop-band area especially), since the ideal filter characteristics in practice cannot be achieved. Thus, various approximations have been created in accordance with the desired transfer properties of the filter [2]. These transfer properties can be specified by the following criteria: (a) the steepness of the transition between the pass-band and stop-band of the magnitude characteristics, (b) the linearity of the phase response, (c) the flatness of the group delay, (d) the ripple of the magnitude characteristics in the pass-band area, (e) the overshot of the step response, and (f) the sensitivity of individual filter parameters in dependence on the selected approximation. Therefore, the proper selection of the approximation is of great importance in the design procedure of a filter as it fundamentally affects the resulting behavior of the filter and consequently the behavior of the whole application [3]. Despite the existence of many various approximations, the selection is usually limited to available standard approximations. In fact, the process of the selection of a suitable approximation in active analog low-frequency (up to hundreds of megahertz) filters, regardless of its importance, is often limited to the selection of the most typical choice (i.e., Butterworth approximation).

Some papers [4,5,6,7,8] have focused on the comparison of multiple approximations in accordance with the above-mentioned criteria in search for the optimal filter design for an intended solution. It is a well-known fact that better characteristics of the magnitude response (the high steepness of the transition between the pass-band and stop-band area) lead to the worse characteristics in the case of the time (the overshoot of the step response) and phase (linearity of the phase response and flatness of the group delay) responses and vice versa. Therefore, we usually look for some compromise or a specific characteristic that is most important for us in a given scenario. We can identify typical frequency filters having Butterworth approximation [9,10,11,12,13], filters having Bessel approximation [14,15,16], filters having Elliptic (also called Cauer) approximation [17,18,19], filters having Chebyshev approximation [20,21,22,23] and filters having Inverse Chebyshev approximation [24,25,26]. Considering the steepness of the transition between the pass-band and stop-band area of the magnitude response, Elliptic and Chebyshev approximations are steepest. Bessel approximation offers the least-steep transition. Opposed to the highest steepness of Elliptic and Chebyshev approximations, these approximations are characterized by the ripple in their pass-band area, unlike Butterworth, Inverse Chebyshev and Bessel approximations. Bessel approximation has the flattest group delay in contrast with the approximations with high steepness (Elliptic and Chebyshev). Butterworth and Inverse Chebyshev approximations show their average properties when it comes to the group delay response. Similarly, Bessel approximation provides the best step-response results, with minimal overshots, at the expense of the steepness of the transition in contrast to Elliptic and Chebyshev approximations. The average transient characteristics, again, are offered by Butterworth and Inverse Chebyshev approximations [3].

From the proposed research [9,10,11,12,13,14,15,16,17,18,19,20,21,22,23,24,25,26], some circuit solutions [12,13,15] present more than one filtering function. However, these functions are typically available from individual circuits (each filter type has its own topology). Similarly, papers [14,16,17,18,25,26] has offered the designs of various orders. However, each filter of a given order was proposed as a single-purpose circuit (each filter was proposed as an individual topology). Only the solution in [12] allows the reconfiguration of the type and order of the resulting filtering function (from a single topology) by the implementation of a switching mechanism. Furthermore, as evident from [9,10,11,12,13,14,15,16,17,18,19,20,21,22,23,24,25,26], the filters typically follow one particular approximation. This approximation then determines the features of the filter and consequently the applicability of the filter in accordance with the desired requirements of a particular application. Thus, the possibility to select or modify (readjust) the approximation of the filter offers a wider range of applicability of the filter and also an additional degree of freedom for the change in the magnitude/phase/transient response.

The above-mentioned issues can be solved by so-called reconnection-less reconfigurable filters [27,28,29,30,31,32,33,34,35,36]. These filters are defined as two-port structures (containing one input and one output terminal), where the resulting transfer function (type) is given by the electronic means. This concept originates from the reconnection-less reconfigurable frequency filters working in microwave systems [27,28,29,30,31,32], where the reconfiguration of the transfer function is achieved by the electromagnetic coupling of elements. From the reconnection-less reconfigurable frequency filters working in the microwave bands, the proposal in [30,31,32] considers the possibility to change the approximation characteristics. The proposed filter in [30] can change between a band-pass function of Chebyshev and Elliptic characteristics. The solution in [31] provides a band-pass function with Butterworth characteristics and a band-stop function with Chebyshev characteristics. The filter in [32] offers a possibility to change between a band-pass function with Chebyshev and Elliptic characteristics. In case of the low-frequency (up to hundreds of megahertz) reconnection-less reconfigurable filters [33,34,35,36], the reconfiguration is performed by the setting of electronically controllable parameters (continuous electronic control) of modern active elements rather than by switching between the inputs and/or outputs of a given filter, or any modification of the internal topology of the filter. Therefore, the resulting transfer function can be changed by the control DC current or voltage externally applied to a chip, instead of the switching or any topological modification of the internal structure (which is typically not possible in the case of on-chip implementation). Moreover, the presence of the continuous electronic control offers a feature of fine-tuning of the resulting function and the possible adjustment of the stop-band/pass-band area. The filters in [33,34,35] provide the ability to change between different types of the transfer function, while the solution in [36] can also change the order (slope between pass and stop band) of the used transfer function.

The practical applications of adjustable and reconfigurable filters can be found in wireless communication and cognitive radio environments, where these filters can be helpful to radio systems in order to isolate signals of interest or attenuate interfering signals depending on the current state of the cognitive environment, as demonstrated in [30,31,32]. These filters usually can change their transfer function between band-pass and band-stop or all-pass functions, or change their approximation type, as mentioned earlier. The reconfigurable filters are also useful in adaptive filtering [37,38,39,40], where automatic adjustment of standard and special transfer responses is welcomed. In addition, the presented solution simplifies ways of design and the overall complexity used, for example, in [37,38]. Adaptive filtering has benefits also for the preprocessing of a signal before analog-to-digital conversion [39], as well as communication systems (interference cancellation) [40].

Table 1 provides a comparison of higher-order (>2) filters referenced in the introduction. This table indicates the following disadvantages of the previously proposed filters: All designs focus only on the utilization of a single particular approximation type (without considering the possibility of changing it).The electronic control of the pole (cut-off) frequency is available (considered) only in papers [21,22,23].None of the designs offer the electronic control (reconnection-less reconfiguration) of the order. Reference [12] offers the change in the order together with the change in the type of the transfer function. Nonetheless, this is performed using mechanical switches.Only research discussed in [9,11,26] is also supported by the experimental measurements.

To the best of the authors’ knowledge, there has been no report of a low-frequency current-mode filter offering the electronic change in the approximation characteristics. The designed filter offers the feature of the electronic change in the used approximation (tested for Butterworth, Bessel, Elliptic, Chebyshev and Inverse Chebyshev approximations) and reconnection-less reconfiguration of the order (providing the low-pass function of the 1st, 2nd, 3rd and 4th order). The electronic adjustment of the cut-off frequency is also possible. The proposal is verified by PSpice simulations together with the experimental measurements of the implemented filter. The further analysis is focused on the comparison of the features of the filter depending on the selected approximation.

## 2. Description of the Filter

The filtering topology, originally proposed in [36] for purposes of the fractional-order current-mode reconnection-less reconfigurable low-pass filter of various orders, was suitably applied to operate as a filter with a reconnection-less reconfiguration of its order and electronic change in used approximation. The filter (Figure 1) was based on the 4th-order leap-frog topology, employing four Operational Transconductance Amplifiers (OTAs) [41], one current amplifier with independently controlled outputs (referred to as Individual Output Gain Controlled Current Amplifier (IOGC–CA)) and four grounded capacitors.

The schematic symbol of the OTA and its used implementation is shown in Figure 2a,b, respectively. The behavior of the OTA element can be described by the relation *I*_OUT±_ = ±*g*_m_ (*V*_IN+_ − *V*_IN–_), where *g*_m_ denotes the transconductance of the OTA. The OTA in the filter structure requires two outputs (of the opposite polarity). Since the commercially available devices, working as the OTA, typically offer only one output, the implemented solution was created by one LT1228 device [42] functioning as the OTA (*g*_m_ tunable by DC control current *I*_SET_gm_) and one EL4083 device [43] (it copies the output current of the OTA and provides two currents of the opposite polarity). Both devices are commercially available. When considering a significant sensitivity of transconductance on temperature variations, external circuits generating the same bias current deviation caused by temperature can be used for auto-compensation of the gm temperature drift of the OTAs when the ambient temperature variation is very large (automotive, military purposes of application). However, based on the application field, the temperature variation expected in standard room conditions has an insignificant impact on the performance of the OTA.

Figure 3 presents the schematic symbol and used implementation of the IOGC–CA element. The IOGC–CA (this particular solution) is described by the relation *I*_OUT_ = −*B*_i_(*I*_IN_), where *B*_i_ stands for the current gain of the individual output of this element; thus, *i* = {1, 2, 3, 4}. The IOGC–CA provides the feature of the reconnection-less reconfiguration of the order of the filter based on the setting of the current gains *B*_i_. It is implemented by one Universal Current Conveyor (UCC) [44], which is used to provide four copies of the input current (it works as a current follower with two inverting and two non-inverting outputs) and four EL2082 devices [45] used to adjust the current gain of each output of the IOGC–CA (controllable by DC control voltage *V*_SET_B_). There are also two additional OPA860 devices [46] (added in order to invert the polarity of the currents so all available transfer functions have the same polarity). Both EL2082 and OPA860 are commercially available. The UCC is not commercially available; nonetheless, it could be implemented by three EL4083 devices or by a suitable CMOS structure, for instance. The inner topology of the IOGC–CA in case of the CMOS structure could be simplified as the outputs of the UCC could be all designed with the same polarity (inversion performed by the OPA860 devices would not be necessary).

The transfer function of the filter (Figure 1) is expressed as *K*(***s***) = *N*(***s***)/*D*(***s***), where
(1)$$\begin{array}{l} N(s)=B_{4}(s^{3}C_{1}C_{2}C_{3}g_{m4}+s^{2}C_{2}C_{3}g_{m1}g_{m4}+sC_{3}g_{m1}g_{m2}g_{m4}+\\ +sC_{1}g_{m2}g_{m3}g_{m4}+g_{m1}g_{m2}g_{m3}g_{m4})+B_{3}(s^{2}C_{1}C_{2}g_{m3}g_{m4}+\\ +sC_{2}g_{m1}g_{m3}g_{m4}+g_{m1}g_{m2}g_{m3}g_{m4})+B_{2}(sC_{1}g_{m2}g_{m3}g_{m4}+\\ +g_{m1}g_{m2}g_{m3}g_{m4})+g_{m1}g_{m2}g_{m3}g_{m4}B_{1} \end{array},$$
(2)$$\begin{array}{l} D(s)=s^{4}C_{1}C_{2}C_{3}C_{4}+s^{3}C_{2}C_{3}C_{4}g_{m1}+s^{2}C_{3}C_{4}g_{m1}g_{m2}+\\ +s^{2}C_{1}C_{4}g_{m2}g_{m3}+s^{2}C_{1}C_{2}g_{m3}g_{m4}+sC_{2}g_{m1}g_{m3}g_{m4}+\\ +sC_{4}g_{m1}g_{m2}g_{m3}+g_{m1}g_{m2}g_{m3}g_{m4} \end{array}.$$

Equation (1) indicates clear dependence of the resulting order of the function on the setting of current gains *B*_1_ to *B*_4_ cancelling corresponding terms of the numerator if a specific current gain is set to zero. For instance, the 4th-order LP function will be available when *B*_1_ = 1, *B*_2_ = *B*_3_ = *B*_4_ = 0. Also, the resulting filtering response can be amplified if particular current gain *B* is higher than 1. Furthermore, the obtainment of the selected approximation characteristics is achieved through the change in the values of the transconductances, dependent on particular coefficients of the transfer function (regarding the chosen approximation). Moreover, if the particular implementation of the current amplifiers offers the analog (continuous) control of their current gain, this feature can be used to adjust the pass-band gain of the output response (fine tuning) if the pass band does not have the unity gain (is not exactly 0 dB due to inaccuracy of filter parameters and values of passive parts).

## 3. General Design Verification

The design was verified using PSpice simulations and experimental measurements. The simulations were performed using models in TSMC 0.18 µm CMOS technology. Used models of OTA, the current amplifier and current follower can be found in [33,47,48]. The transconductance of the OTA model and the current gain of the current amplifier model were adjusted using a DC control current. The supply voltage of all used models was ±1 V.

The practical implementation of individual active elements was performed by the UCC, EL2082, EL4083, LT1228 and OPA860, as introduced in the previous section. The UCC, made by Brno University of Technology, and the ON Semiconductor design center in I3T 0.35 µm CMOS technology, used a supply voltage of ±1.65 V. The remaining used active elements used a supply voltage of ±5 V. The measurement itself was performed by a network analyzer Agilent 4395A utilizing voltage-to-current and current-to-voltage converters constructed by OPA860 and OPA861 [49] devices. Figure 4 shows the implemented PCB of the used filter.

The design of the filters of a higher order (>2) is usually performed by the cascade combination of individual 1st- and 2nd-order filters. For the direct design, coefficients of the transfer function (coefficients *b*), depending on the chosen order and approximation, have to be calculated [3]. In our case, the coefficients were calculated using a design tool NAF [50] (or they can be obtained from tables with normalized coefficients of the transfer function as in [3], for instance). The general 4th-order transfer function had the form given by (3). The terms contained in the numerator may vary (based on filter type), as long as the order of the highest polynomial term(s) is not higher than the order of the highest polynomial in the denominator in order for the circuit to be stable. Coefficients *b* can then be obtained based on chosen parameters such as the angular frequency, approximation, steepness of the transition, etc., (the tolerance field) from design tables or NAF.
(3)$$K(s)=\frac{N(s)}{D(s)}=\frac{N(s)}{b_{0}+b_{1}s^{1}+b_{2}s^{2}+b_{3}s^{3}+b_{4}s^{4}}$$

The following specification of the tolerance field for the calculation of the coefficients, was used: approximation—(gradually) Elliptic, Butterworth, Chebyshev, Bessel and Inverse Chebyshev, operational angular frequency, 300,000 rad/s (*f*_0_ ≈ 47 kHz) (this frequency was chosen considering the bandwidth limitations of the used active elements (5–10 MHz) so there were at least two decades before the response was affected by parasitic characteristics); ripple in the pass band (if any)—*K*_P_ = 3 dB, stop-band frequency *f*_S_—3,000,000 rad/s (470 kHz); transfer in stop band (*K*_S_)—depending on used approximation in order to create the transfer function of a given order (4th-order) −84 dB (Elliptic), −77 dB (Butterworth), −73 dB (Chebyshev), −65 dB (Bessel), and −73 dB (Inverse Chebyshev). Based on the selected approximation, the coefficients of the transfer function were calculated, as stated in Table 2.

The relations for the transconductances *g_m_*_1_ to *g_m_*_4_ can be expressed by the comparison of individual terms of the denominator of the transfer function of the filter (2) and the general transfer function of the 4th-order (3):(4)$$g_{m1}=b_{3}C_{1},$$
(5)$$g_{m2}=\frac{b_{2}C_{1}C_{2}C_{3}C_{4}-C_{1}C_{2}g_{m3}g_{m4}}{C_{1}C_{4}g_{m3}+C_{3}C_{4}g_{m1}},$$
(6)$$g_{m3}=-\frac{C_{1}C_{3}C_{4}b_{1}g_{m1}}{C_{1}^{2}C_{4}b_{1}-C_{1}C_{4}b_{2}g_{m1}-g_{m}{}_{1}^{2}g_{m4}},$$
(7)$$g_{m4}=\frac{C_{1}C_{4}b_{0}(C_{1}b_{1}-b_{2}g_{m1})}{C_{1}^{2}b_{1}^{2}-C_{1}b_{1}b_{2}g_{m1}+b_{0}g_{m}{}_{1}^{2}}.$$

If choosing the values of capacitors *C*_1_ = *C*_2_ = *C*_3_ = *C*_4_ = 1 nF, the resulting values of the transconductances (based on coefficients from Table 2) are given in Table 3.

Figure 5, Figure 6, Figure 7, Figure 8 and Figure 9 show the results (simulations denoted by black dashed lines and experimental measurements presented by colored lines) of the output response of the filter for all available orders gradually with Elliptic approximation characteristics (Figure 5), Butterworth approximation characteristics (Figure 6), Chebyshev approximation characteristics (Figure 7), Bessel approximation characteristics (Figure 8) and Inverse Chebyshev approximation characteristics (Figure 9). The results show expected features of given approximations, such as the ripple in the pass-band area, in the case of responses with Elliptic and Chebyshev approximation, the maximally flat pass-band area for Butterworth, Bessel and Inverse Chebyshev approximations, a less steep transition between the pass-band and stop-band area in case of Bessel approximation, etc. The differences between the simulation results and the experimental results (applies for all presented data in the paper) were mainly caused by the parasitic/real characteristics of used active elements.

The magnitude and phase characteristics (simulations denoted by black dashed lines and experimental measurements presented by colored lines) of the 4th-order (as the highest order responses available in case of the proposal—the best example) output response are shown in Figure 10, for a more direct comparison of the resulting steepness depending on the used approximation. It is evident that the responses with Elliptic and Chebyshev approximation characteristics had a greater steepness of their transition slopes, the Butterworth and Inverse Chebyshev approximations provided average steepness, and the smallest steepness of the magnitude response was obtained for Bessel approximation. The results prove the intended feature of the reconnection-less reconfiguration of the used approximation (electronic change in approximation). We can choose a better fitting slope of the response based on the selected approximation if this characteristic is important for our needs. The pole frequency for each approximation can be compared in Table 4. The pole frequency slightly varied from 42.9 kHz to 49.6 kHz in the cases of the simulations and 43.8 kHz to 47.2 kHz in the case of the measurement. The pole frequency of Bessel approximation was at lower value in comparison to other approximations and Elliptic approximation was at a higher frequency in comparison to other approximations. The transfer in the stop band *K*_S_ at the stop-band frequency *f*_S_ (which was specified to be 470 kHz during the obtainment of the coefficients of the transfer function) can be compared in Table 5. The Elliptic and Chebyshev approximations showed the highest attenuation at this frequency. The smallest attenuation was obtained in the case of Bessel approximation. In order to highlight the faster/slower transition between the pass-band and stop-band area, the frequency of the attenuation reaching −60 dB (when considering the stop band to be for the signal being 1000 times smaller) is summarized in Table 6. Chebyshev and Elliptic approximation reached this attenuation earlier (at frequency around 175 kHz). They were followed by Butterworth and Inv. The Chebyshev approximation (around 265 kHz) and Bessel approximation did not reach this attenuation until 390 kHz. Table 7 provides the information about the ripple (its peak value) in the pass band. There was no ripple in the case of the Butterworth, Bessel and Inv. Chebyshev approximations. Elliptic and Chebyshev approximations showed similar peak values of their ripples, with the ripple being more evident in the case of the measurement.

## 4. Further Analysis and Comparison

The simulation and experimental results provided in this section were carried out using the same setup as described in the previous section (unless stated otherwise).

### 4.1. Electronic Adjustment of the Cut-off Frequency

The electronic adjustment of the cut-off frequency was achieved by the change in the values of transconductances *g_m_*_1_ to *g_m_*_4_, as long as the ratio between them remained unchanged so it did not affect the approximation features and the value of the quality factor. This was tested in case of the 4th-order function with Butterworth characteristics for three settings of the desired (theoretical) *f*_0_ (23.5 kHz, 47 kHz and 94 kHz). The resulting values of the transconductances in relation to the selected value of *f*_0_ are summarized in Table 8. The results of the electronic adjustment of the cut-off frequency are depicted in Figure 11 and compared in Table 9. The intended electronic adjustment operated as expected without an influence on the characteristics of the used approximation or the value of the quality factor.

### 4.2. Response to the Square Waveform

The behavior of the filter in the time domain depending on the used approximation was evaluated for the square waveform input signal in order to investigate the overshot of the step response. The results were provided only in the case of the simulations, as the current responses (for the measurements) would be rather difficult to acquire without proper equipment. The input (square) signal had its amplitude set to 15 µA with a frequency of 1 kHz. Figure 12 presents the output responses (colored traces) to the input signal (black trace), including a detail of the overshot in case of the rising edge. The filter worked in the time domain as expected, depending on the configuration of the reconnection-less reconfiguration of a given approximation. The Elliptic approximation had the greatest overshot in response to the unity step followed by Chebyshev approximation. The response with Bessel approximation characteristics had the most flat unity step response with the smallest overshot. Table 10 compares the settling time based on the approximation. Bessel approximation showed the shortest settling time, as expected. The settling times of Butterworth and Inverse Chebyshev approximations were close to each other, and they are slightly longer than in case of Bessel approximation. Chebyshev and Elliptic approximations show the most significant settling times from all tested approximations, with Elliptic approximation as the longest. This behavior follows features expected from magnitude and phase characteristics. By electronic means, we can choose approximation characteristics with a smaller overshot if the application is more sensitive to stability issues.

### 4.3. Group Delay-Response Results

Since the group delay characteristics are associated with the linearity of the phase responses and analogically with the step response results, the response with Bessel approximation was the one with a minimum deviation from the constant group delay time in the pass-band area. Furthermore, Elliptic and Chebyshev approximation showed the worst group delay properties from used approximations, and Butterworth and Inverse Chebyshev approximations offered relatively small ripples in the group delay responses to the pass-band area, as demonstrated by Figure 13. The peak values of the group delay of individual approximations can be compared in Table 11. The Bessel showed no ripple, with the highest value of 0.7 µs in the case of the simulations and 0.8 µs for the measurement. The peak value of Butterworth and Inv. Chebyshev approximations reached around 1.3 µs. The Elliptic approximation showed a lower ripple than the Chebyshev approximation, which exhibited the largest ripple in group delay. The group delay responses were provided for the 4th-order function.

### 4.4. Sensitivity Analysis

A certain inaccuracy (impedance, offset, etc.) of individual outputs of used active elements, together with the tolerance of used passive parts, can cause a significant deviation in the resulting transfer function. Since each approximation will lead to different values of used transconductances and thus, it might lead (depending on the implementation) to different values of output impedances of used active elements interacting with each other; the sensitivity of individual parameters of the filter will vary depending on the currently used approximation. Therefore, an analysis of the subsequent sensitivity functions in relation to the used approximation can prove useful. Based on the above-mentioned, the real transfer function of the filter consists of 16 individual parameters (*C*_1_, *C*_2_, *C*_3_, *C*_4_, *g_m_*_11_, *g_m_*_12_, *g_m_*_21_, *g_m_*_22_, *g_m_*_31_, *g_m_*_31_, *g_m_*_41_, *g_m_*_42_, *B*_1_, *B*_2_, *B*_3_ and *B*_4_). Parameters *g_m_*_11_, *g_m_*_12_, *g_m_*_21_, *g_m_*_22_, *g_m_*_31_, *g_m_*_31_, *g_m_*_41_, *g_m_*_42_ represent individual outputs of OTAs and parameters *B*_1_, *B*_2_, *B*_3_ and *B*_4_ then stand for the outputs of the IOGC–CA element. Considering all these parameters, the real transfer function takes form of *K*_real_(***s***) = *N*_real_(***s***)/*D*_real_(***s***), where
(8)$$\begin{array}{cc} \begin{array}{l} N_{real}(s)=s^{3}C_{1}C_{2}C_{3}g_{m42}B_{4}+\\ +s^{2}(C_{1}C_{2}g_{m32}g_{m42}B_{3}+C_{2}C_{3}g_{m11}g_{m42}B_{4})+\\ +s(C_{1}g_{m22}g_{m32}g_{m42}B_{2}+C_{2}g_{m11}g_{m32}g_{m42}B_{3}+\\ +C_{3}g_{m12}g_{m21}g_{m42}B_{4}+C_{1}g_{m22}g_{m31}g_{m42}B_{4})+\\ +g_{m12}g_{m22}g_{m32}g_{m42}B_{1}+g_{m11}g_{m22}g_{m32}g_{m42}B_{2}+\\ +g_{m12}g_{m21}g_{m32}g_{m42}B_{3}+g_{m11}g_{m22}g_{m31}g_{m42}B_{4} \end{array} & , \end{array}$$
(9)$$\begin{array}{cc} \begin{array}{l} D_{real}(s)=s^{4}C_{1}C_{2}C_{3}C_{4}+\\ +s^{3}C_{2}C_{3}C_{4}g_{m11}+s^{2}(C_{1}C_{4}g_{m22}g_{m31}+\\ +C_{1}C_{2}g_{m32}g_{m41}+C_{3}C_{4}g_{m12}g_{m21})+\\ +s(C_{4}g_{m11}g_{m22}g_{m31}+C_{2}g_{m11}g_{m32}g_{m41})+\\ +g_{m12}g_{m21}g_{m32}g_{m41} \end{array} & . \end{array}$$

The relative sensitivity function (magnitude) of the filter in relation to the variation of the individual parameters can be described as [51]
(10)Sqi|K(jω)|=Re{Sqi|K(jω)|}
where ***K***(*jω*) is the transfer of the filter and *q*_i_ represents *i*th parameter of the filter. The mathematical expression of individual sensitivities was calculated using Maple software.

Figure 14, Figure 15, Figure 16, Figure 17 and Figure 18 depict the relative sensitivities of individual parameters within a whole analyzed frequency range (100 Hz to 100 MHz) for the given approximations. The sensitivity analysis of the complete transfer function (all current gains *B* were set to one) is performed so it includes the results for all parameters. It can be seen that the sensitivities reached typical values (sensitivity around 1), with the highest values around the designated cut-off frequency (*f*_0_ = 47 kHz). Elliptic and Chebyshev approximations showed higher sensitivities (the most sensitive parameters in general were *C*_1_ and *C*_3_ reaching over 2 and −3 around *f*_0_). It is evident that the sensitivities for Butterworth, Bessel and Inverse Chebyshev approximations are slightly lower.

### 4.5. General Comparison of the Values of Resulting Transconductances

All used values of transconductances (regarding selected approximation) are summarized in Table 3. It can be seen that Elliptic and Chebyshev approximations require lower values of the transconductances than the remaining approximations. The highest value of used transconductances for a given setting (*f*_0_ = 47 kHz and selected values of capacitors) for Elliptic approximation was 342.6 µs and was 283.6 µs in the case of Chebyshev approximation. Butterworth and Inverse Chebyshev characteristics required similar values of transconductances (the highest value for Butterworth was 784.4 µs and was 789.5 µs for Inverse Chebyshev approximation, which was approximately two times larger than in the case of Elliptic and Chebyshev approximations). Bessel approximation was characterized by the highest required values of the transconductances (the highest used value was 1421.4 µs, which was double the value of transconductance used in Butterworth and Inverse Chebyshev approximations and four times the value required for Elliptic and Chebyshev approximations). Considering typical implementation limits and the related fabrication issues of CMOS structures and commercially available solutions of the OTA element, the maximal (usually) obtainable value of the transconductance was around 1 ms for modern integrated CMOS designs and 10–50 ms for BJT commercially available devices. Therefore, from the implementation aspects, Elliptic and Chebyshev approximations came out as more advantageous ones. Moreover, we need to keep in mind that, in case of the adjustment of the cut-off frequency, the value of transconductances is twice as much if the value of cut-off frequency is doubled (see Table 8). This fact creates more restricted limits of the cut-off-frequency adjustment in cases of approximations with higher required values of resulting transconductances.

## 5. Filter Utilization

The filter has to be able to process different types of signals from simple shapes to more complex ones. In this case, various approximations can be found more suitable for specific types of signals. Therefore, in order to point out a beneficial feature of the electronic change in the approximation, the following situation is considered: the proposed filter will be tested for two types of processed input signal. The first signal is of a simple shape (sine–wave signal with a low number of spectral components (if slightly distorted/not ideal)), in comparison to the second signal, having more complex shape (ramp signal consisting of multiples and combinations of spectral components). Furthermore, the processed (useful) signal is affected by noise. Figure 19 depicts the time-domain representation and spectrum of the ramp signal (of a frequency of 20 kHz and an amplitude equal to 200 mV peak-to-peak) together with a signal affected by noise (1 V peak-to-peak and a bandwidth of 10 MHz). The time-domain waveform and spectrum of the sine signal (of a frequency of 20 kHz and an amplitude equal to 150 mV peak-to-peak), together with a signal affected by noise (with the same parameters as in previous case), is presented in Figure 20.

For the purposes of the demonstration, the 4th-order filtering function (*f*_0_ = 47 kHz) was used with Bessel, Butterworth and Elliptic approximations chosen as representative examples. The transfer (attenuation) of the used V/I, I/V converters was approximately –12 dB. Figure 21, Figure 22 and Figure 23 subsequently show the output (filtrated) signal and its spectrum when the electronic change in the approximation was set to Bessel, Butterworth and Elliptic approximation characteristics. When the Butterworth and Elliptic approximation was used in order to process the ramp signal, it can be seen that the filtered signal had a significant deformation of its shape. This was caused by the fact that the transition between the pass-band and stop-band area of the function with Butterworth and Elliptic approximation characteristics was steeper than for the Bessel approximation, resulting in higher spectral components of the useful signal being partially filtered out as well. Therefore, for the more complex signal (shape), we should consider approximations with a less steep transition between the pass-band and stop-band area. On the other hand, when processing a simple signal (sine waveform in this case), we can select an approximation with a steeper transition between pass-band and stop-band area for better filtration of the noise without the signal shape deformations. This could be, of course, solved by changing the order of the filter. Nonetheless, it means more additional stages of the building blocks (integrators) in a cascade when a topological modification (increasing circuit complexity if not considering the reconnection-less reconfiguration) would be necessary. Extended complexity would also mean additional power consumption as well as additional chip area. When using the electronic change in the approximation characteristics, the steepness of the function can be easily adapted for the specific type of the processed signal.

## 6. Conclusions

The proposed filter offers the electronic change in the approximation characteristics, reconnection-less reconfiguration of the order and the electronic adjustment of the cut-off frequency. The performance of this filter was verified through PSpice simulations and experimental measurements, proving the presence and intended function of all above-mentioned features (see Figure 5, Figure 6, Figure 7, Figure 8, Figure 9, Figure 10, Figure 11, Figure 12 and Figure 13 and corresponding text).

The advantages of the presented filter are the following:The reconnection-less reconfiguration of the order (higher-order filters of different orders are usually designed as individual circuits or the solution contains electronic switches—the disadvantages of electronic switches was discussed in the introduction);The ability to electronically change approximation characteristics;The ability of fine tuning (adjusting the pass-band area);The electronic adjustment of the pole frequency.

The above-mentioned features provide additional degrees of freedom for the filter and its adjustment based on the application. With these features, the filter can adapt to a changing situation and requirements such as sensors, wireless communication and cognitive radio environments, where a change in the filtering function might be necessary. Standard filtering approaches and multifunctional filters do not allow these features.

The electronic change in the approximation characteristics can be useful in order to influence the resulting behavior of the filter when searching for the optimal characteristics for intended application or based on the processed signal. Based on this fact, the proposed filter offers the possibility to choose the most fitting characteristics when focusing on a particular feature (magnitude characteristics, transient domain features, implementation requirements and limitations, etc.). For instance, the steepness of the transition between the pass-band area and stop-band area can be adjusted without having to add another integrator in a cascade, as topology modification might not be possible (in the case of on-chip implementation). As another example, the approximation with smaller overshots of the step response can be used for applications which are more sensitive when it comes to their stability. All this can be achieved from one topology that offers the electronic adjustment of its approximation characteristics, which can be more freely adjusted for particular needs of a given application, or that can be adjusted anytime during the lifespan if the parameters of the application change or deteriorate. Another possible use of this ability to change the approximation characteristics of the filter could be served in case of sensors, where the conditions for the signal change often and the ability to change the approximation characteristics can be used depending on the current situation in order to decrease the noise in the processed signal. Particular benefits of the electronic change in the approximation characteristics were discussed and an example was presented (see Figure 21, Figure 22 and Figure 23).

## Figures and Tables

**Figure 1 sensors-23-08057-f001:**
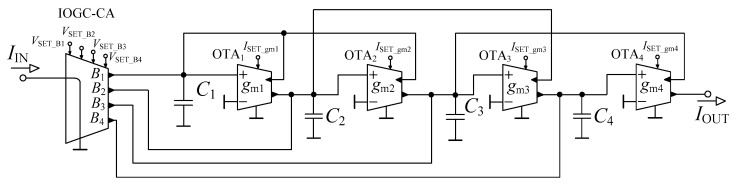
Designed low-pass filter with the ability of the reconnection-less reconfiguration of its order and used approximation based on the 4th-order leap-frog topology.

**Figure 2 sensors-23-08057-f002:**
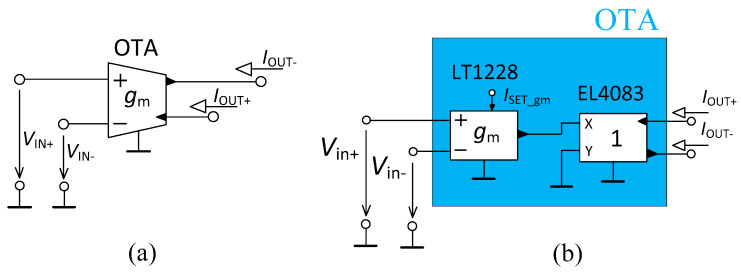
Operational Transconductance Amplifier (OTA): (**a**) schematic symbol, (**b**) used implementation.

**Figure 3 sensors-23-08057-f003:**
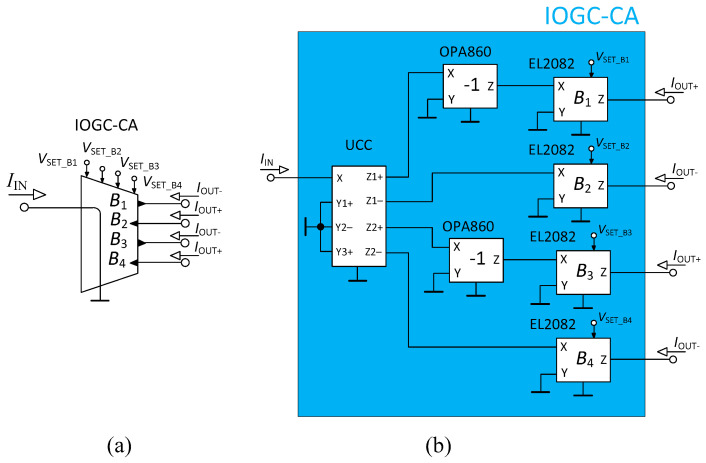
Individual Output Gain Controlled Current Amplifier (IOGC–CA): (**a**) schematic symbol, and (**b**) used implementation.

**Figure 4 sensors-23-08057-f004:**
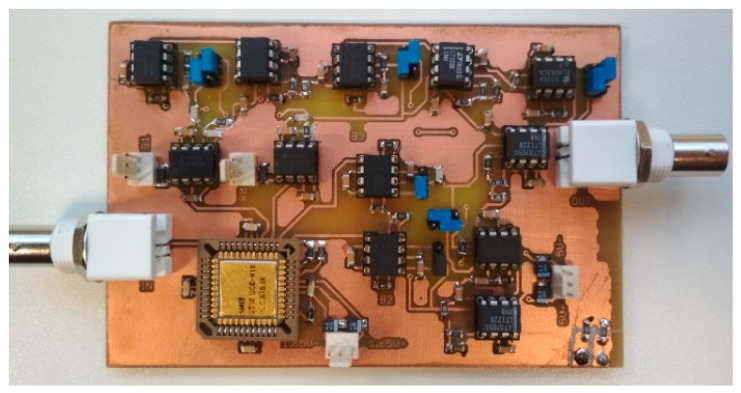
PCB of the implemented filter.

**Figure 5 sensors-23-08057-f005:**
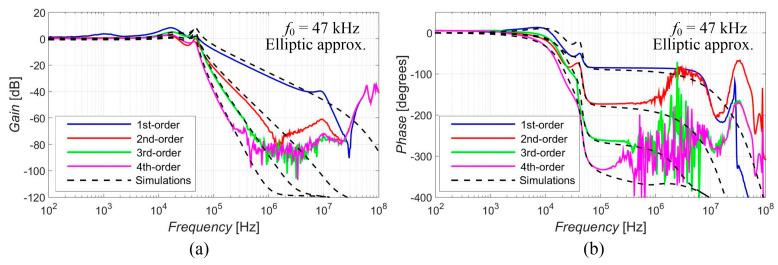
Magnitude (**a**) and phase (**b**) characteristics of all available orders (1st, 2nd, 3rd and 4th) of the output response with Elliptic approximation: simulations (black dashed lines), and experimental results (colored lines).

**Figure 6 sensors-23-08057-f006:**
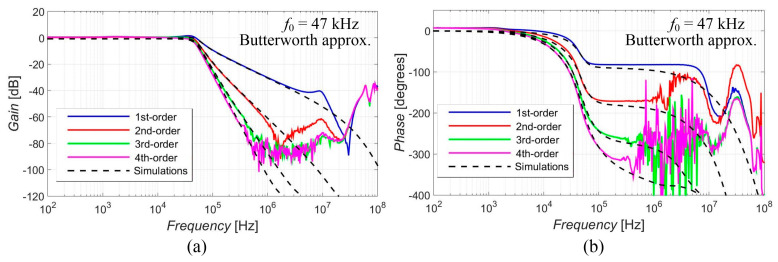
Magnitude (**a**) and phase (**b**) characteristics of all available orders (1st, 2nd, 3rd and 4th) of the output response with Butterworth approximation: simulations (black dashed lines), and experimental results (colored lines).

**Figure 7 sensors-23-08057-f007:**
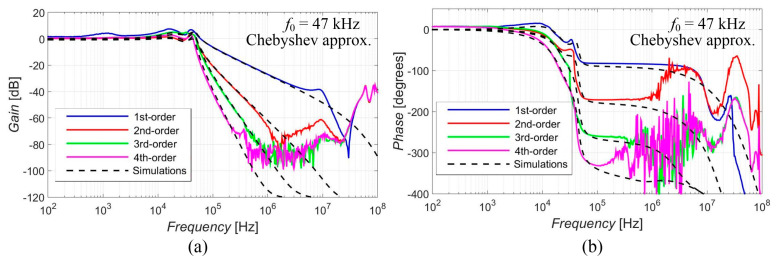
Magnitude (**a**) and phase (**b**) characteristics of all available orders (1st, 2nd, 3rd and 4th) of the output response with Chebyshev approximation: simulations (black dashed lines), and experimental results (colored lines).

**Figure 8 sensors-23-08057-f008:**
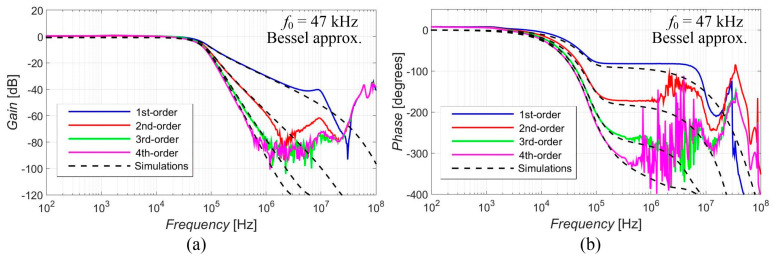
Magnitude (**a**) and phase (**b**) characteristics of all available orders (1st, 2nd, 3rd and 4th) of the output response with Bessel approximation: simulations (black dashed lines), and experimental results (colored lines).

**Figure 9 sensors-23-08057-f009:**
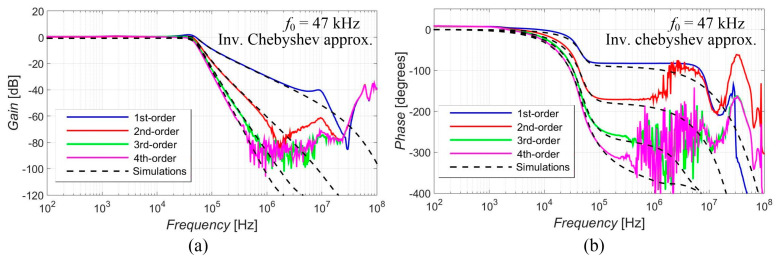
Magnitude (**a**) and phase (**b**) characteristics of all available orders (1st, 2nd, 3rd and 4th) of the output response with Inv. Chebyshev approximation: simulations (black dashed lines), and experimental results (colored lines).

**Figure 10 sensors-23-08057-f010:**
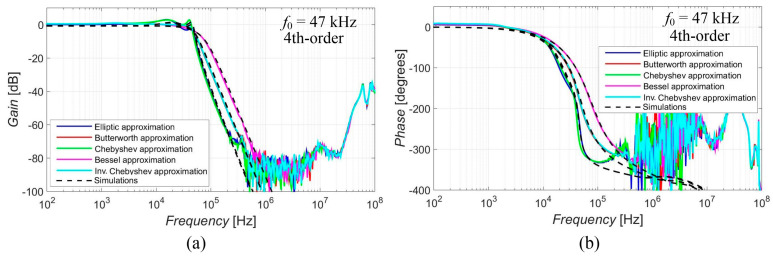
Comparison of the magnitude (**a**) and phase (**b**) 4th-order responses of used approximation.

**Figure 11 sensors-23-08057-f011:**
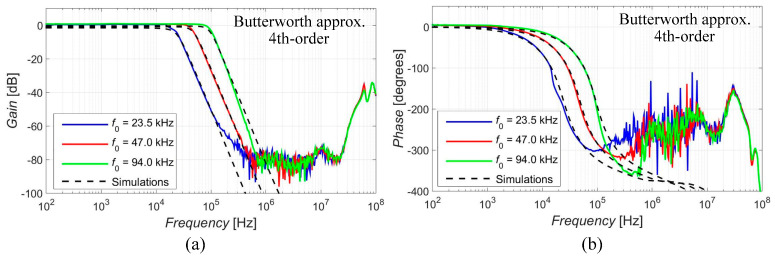
Feature of the electronic adjustment of *f*_0_ (magnitude (**a**) and phase (**b**) characteristics) for three different settings in case of the 4th-order response with Butterworth characteristics.

**Figure 12 sensors-23-08057-f012:**
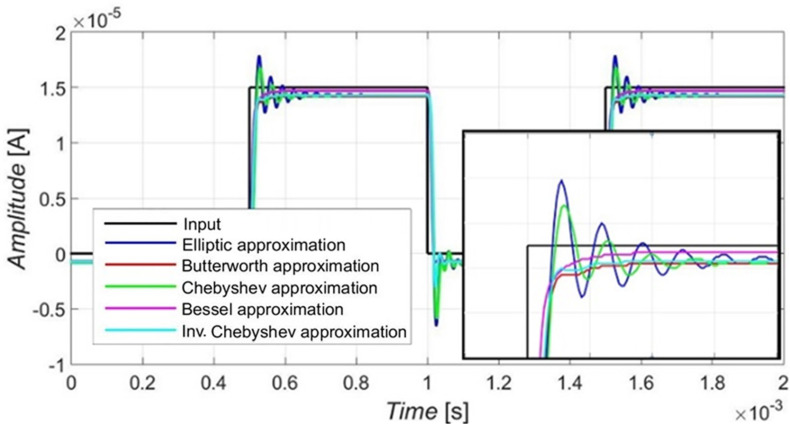
Transient domain response onto a square waveform input signal of 15 µA amplitude and a frequency of 1 kHz.

**Figure 13 sensors-23-08057-f013:**
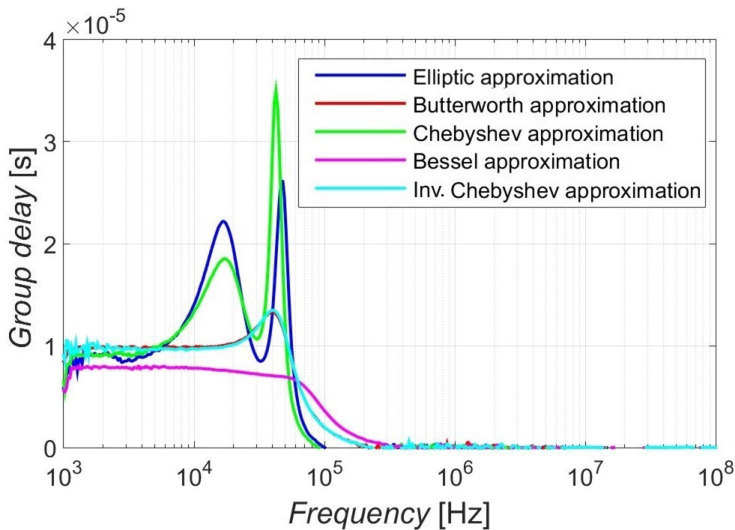
Comparison of the experimental measurements of the group delay responses of used approximations.

**Figure 14 sensors-23-08057-f014:**
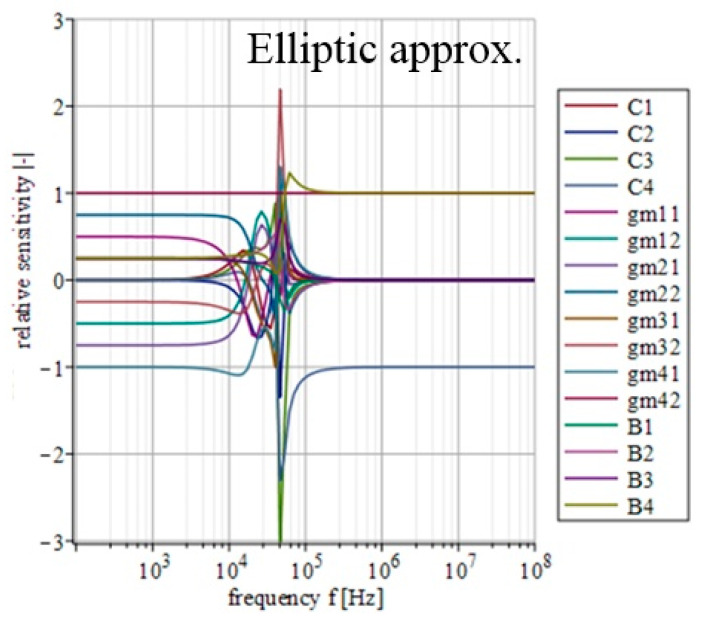
Relative sensitivity functions of magnitude response on individual filter parameters of the transfer function with Elliptic approximation characteristics.

**Figure 15 sensors-23-08057-f015:**
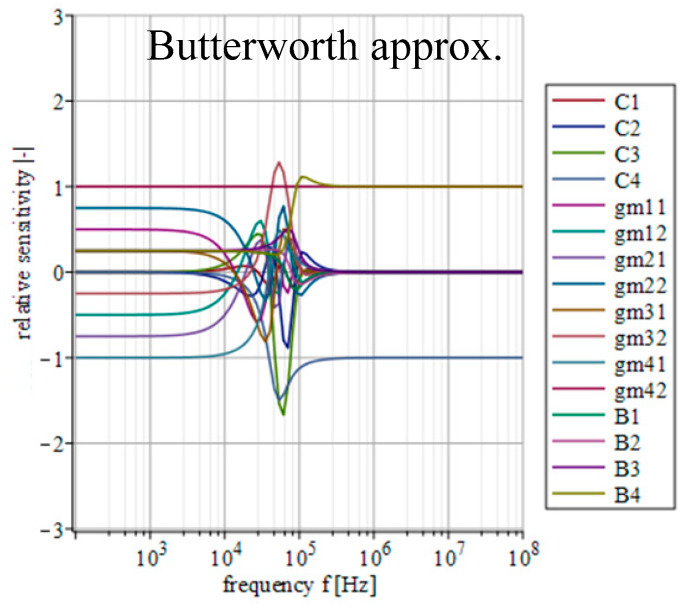
Relative sensitivity functions of magnitude response on individual filter parameters of the transfer function with Butterworth approximation characteristics.

**Figure 16 sensors-23-08057-f016:**
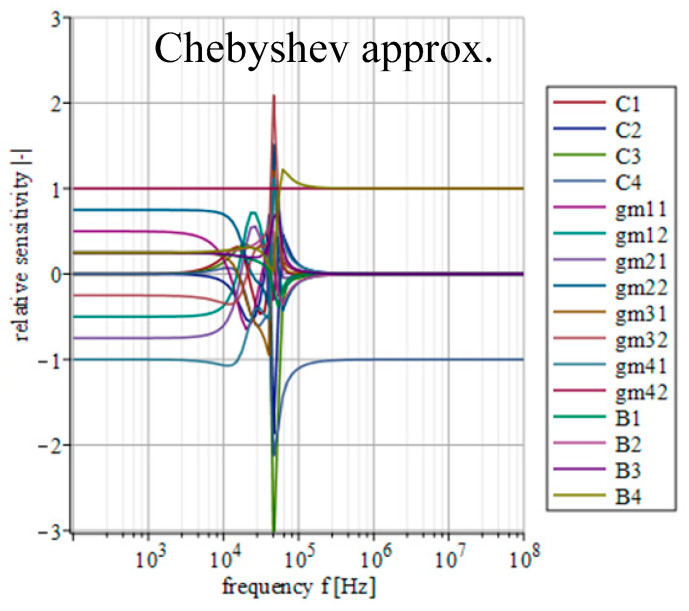
Relative sensitivity functions of magnitude response on individual filter parameters of the transfer function with Chebyshev approximation characteristics.

**Figure 17 sensors-23-08057-f017:**
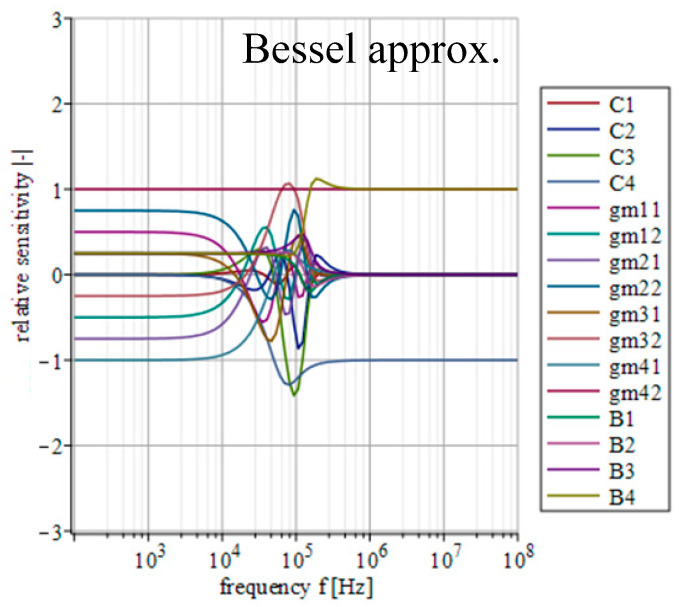
Relative sensitivity functions of magnitude response on individual filter parameters of the transfer function with Bessel approximation characteristics.

**Figure 18 sensors-23-08057-f018:**
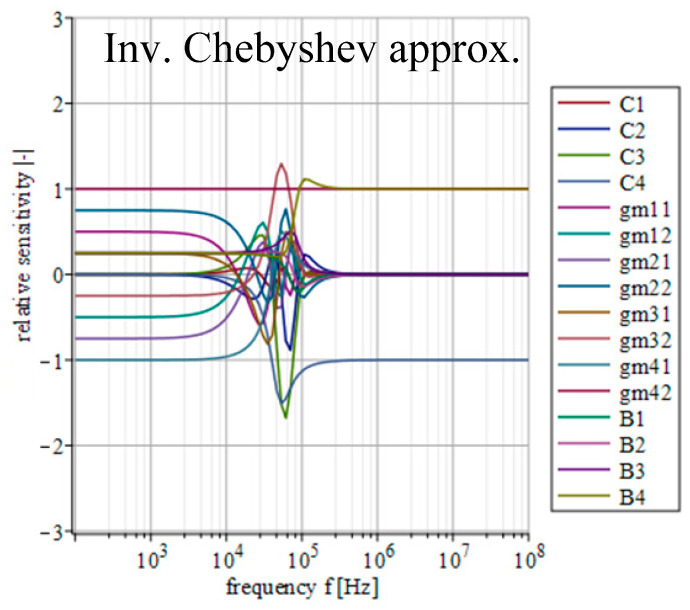
Relative sensitivity functions of magnitude response on individual filter parameters of the transfer function with Inv. Chebyshev approximation characteristics.

**Figure 19 sensors-23-08057-f019:**
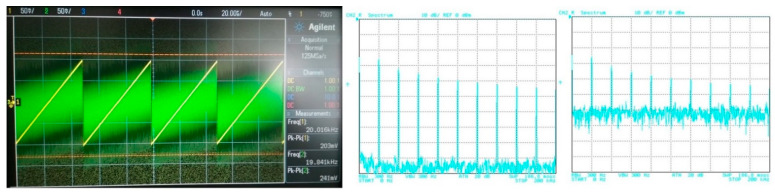
Waveforms of the useful ramp signal (yellow wave) and signal affected by noise (green wave) and their spectrums.

**Figure 20 sensors-23-08057-f020:**
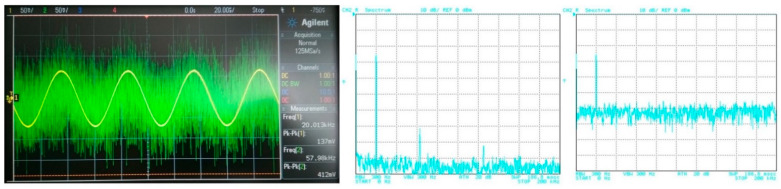
Waveforms of the useful sine signal (yellow wave) and signal affected by noise (green wave) and their spectrums.

**Figure 21 sensors-23-08057-f021:**
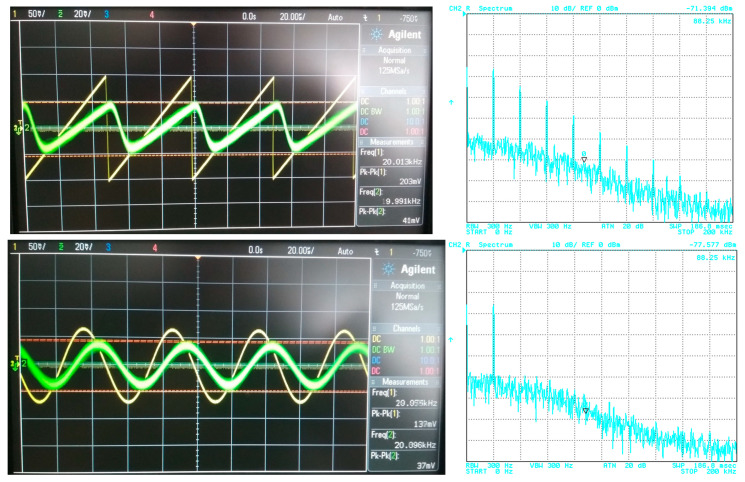
Input (before being affected by noise) and output filtrated signal and resulting output spectrum for Bessel approximation.

**Figure 22 sensors-23-08057-f022:**
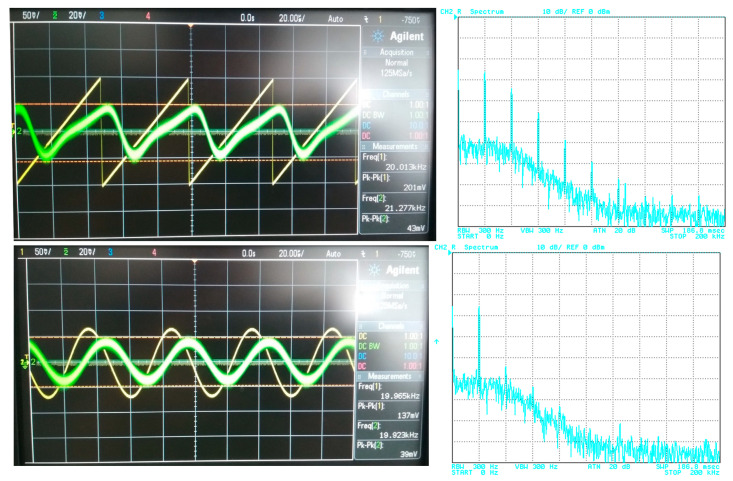
Input (before being affected by noise) and output filtrated signal and resulting output spectrum for Butterworth approximation.

**Figure 23 sensors-23-08057-f023:**
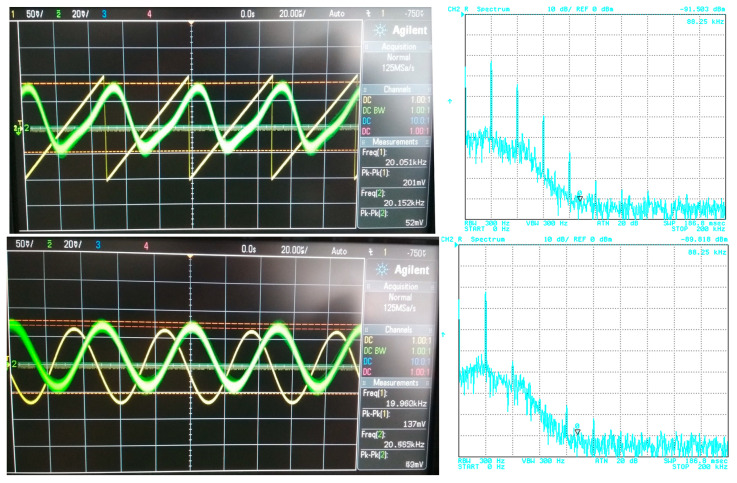
Input (before being affected by noise) and output filtrated signal and resulting output spectrum for Elliptic approximation.

**Table 1 sensors-23-08057-t001:** Comparative summary of cited higher-order low-frequency filter designs.

Reference Number	Year of Publication	Type of Approximation	Type of the Transfer Function	Order	Number of Active/Passive Parts	Simulated/Measured	Operational Mode	Electronic Adjustment of the Pole (Cut-Off) Frequency	Electronic Adjustment of the Order	Electronic Adjustment of the Approximation Type	Notes
[9]	2019	Butterworth	LP	5th	6/5	Yes/Yes	VM	No	No	No	^1,2^
[10]	2018	Butterworth	BP	5th	5/35	Yes/No	CM	No	No	No	^1,2^
[11]	2018	Butterworth	LP	5th	6/5	Yes/Yes	VM	No	No	No	^1,2^
[12]	2016	Butterworth	LP, BP	1st, 2nd, 3rd	3/40	Yes/No	VM	No	No	No	^1,2,3^
[13]	2017	Butterworth	LP, BP	3rd	N/A	Yes/No	CM	No	No	No	^2,4^
[14]	2020	Bessel	LP	2nd–8th	N/A	Yes/No	N/A	No	No	No	^2,5^
[16]	2020	Bessel	LP	4th	N/A	Yes/No	N/A	No	No	No	^2,5^
[17]	2018	Elliptic	LP	3rd–5th	–	Yes/No	VM	No	No	No	^2,5,6^
[18]	2013	Elliptic	LP	3rd, 4th	–	Yes/No	VM	No	No	No	^2,7^
[20]	2019	Chebyshev	BP	6th	6/6	Yes/No	VM	No	No	No	^1,2^
[21]	2018	Chebyshev	LP	4th	4/6	Yes/No	VM	Yes	No	No	–
[22]	2021	Chebyshev	BS	3rd	9/6	Yes/No	CM	Yes	No	No	–
[23]	2020	Chebyshev	HP	3rd	6/3	Yes/No	CM	Yes	No	No	–
[25]	2016	Inv. Chebyshev	BP	3rd	N/A	Yes/No	VM	No	No	No	^8^
[26]	2017	Inv. Chebyshev	LP	3rd	N/A	Yes/Yes	VM	No	No	No	^8^
This work	–	Butterwoth, Bessel, Elliptic, Chebyshev, Inv. Chebyshev	LP	1st, 2nd, 3rd, 4th	5/4	Yes/Yes	CM	Yes	Yes	Yes	–

List of previously unexplained abbreviations used in Table 1: LP—low-pass, BP—band pass, BS—band stop, HP—high-pass, VM—voltage mode, CM—current mode. Notes: ^1^ presented topology is fully differential, ^2^ electronic control of the pole (cut-off) frequency is not available or it is possible but not investigated and presented, ^3^ structure can be reconfigured (change its order) by an array of switches, ^4^ design is CMOS-only, ^5^ simulations of the presented research are strictly numerical, ^6^ number of active/passive parts, depending on the order, is 5/6, 12/8, 15/8, ^7^ number of active passive parts, depending on the order, is 3/10, 4/14 or 5/16, ^8^ presented filter is passive.

**Table 2 sensors-23-08057-t002:** Values of the coefficients of the transfer function for different approximations.

Approximation	Elliptic	Butterworth	Chebyshev	Bessel	Inv. Chebyshev
*b*_4_ [–]	1				
*b*_3_ [–]	1.736 × 10^5^	7.844 × 10^5^	1.745 × 10^5^	1.421 × 10^6^	7.895 × 10^5^
*b*_2_ [–]	1.079 × 10^11^	3.076 × 10^11^	1.052 × 10^11^	9.091 × 10^11^	3.117 × 10^11^
*b*_1_ [–]	1.143 × 10^16^	7.068 × 10^16^	1.093 × 10^16^	3.015 × 10^17^	7.226 × 10^16^
*b*_0_ [–]	1.622 × 10^21^	8.119 × 10^21^	1.434 × 10^21^	4.286 × 10^22^	8.489 × 10^21^

**Table 3 sensors-23-08057-t003:** Values of the transconductances with regard to the selected approximation.

Approximation	Elliptic	Butterworth	Chebyshev	Bessel	Inv. Chebyshev
*g_m_*_1_ [µs]	173.6	784.4	174.5	1421.4	789.5
*g_m_*_2_ [µs]	242.2	277.3	244.1	490.7	278.9
*g_m_*_3_ [µs]	112.6	190.3	118.7	307.4	189.9
*g_m_*_4_ [µs]	342.6	196.1	283.6	200.1	203.0

**Table 4 sensors-23-08057-t004:** The pole frequency *f*_0_ of the 4th-order LP function from Figure 10.

Approximation	Simulations	Measurement
Elliptic	49.6 kHz	47.2 kHz
Butterworth	44.1 kHz	45.2 kHz
Chebyshev	47.8 kHz	45.8 kHz
Bessel	42.9 kHz	43.8 kHz
Inv. Chebyshev	45.3 kHz	45.9 kHz

**Table 5 sensors-23-08057-t005:** The transfer in stop band *K*_S_ at the stop-band frequency *f*_S_ (470 kHz) of the 4th-order LP function from Figure 10.

Approximation	Simulations	Measurement
Elliptic	−92.9 dB	−97.6 dB
Butterworth	−79.4 dB	−77.8 dB
Chebyshev	−94.0 dB	−84.8 dB
Bessel	−65.2 dB	−68.0 dB
Inv. Chebyshev	79.1 dB	−75.4 dB

**Table 6 sensors-23-08057-t006:** The stop-band frequency *f*_S_ if the transfer in stop band *K*_S_ = −60 dB of the 4th-order LP function from Figure 10.

Approximation	Simulations	Measurement
Elliptic	184 kHz	174 kHz
Butterworth	268 kHz	258 kHz
Chebyshev	177 kHz	170 kHz
Bessel	405 kHz	390 kHz
Inv. Chebyshev	270 kHz	257 kHz

**Table 7 sensors-23-08057-t007:** The ripple in the pass band (peak value) of the 4th-order LP function from Figure 10.

Approximation	Simulations	Measurement
Elliptic	1.5 dB	3.1 dB
Butterworth	-	-
Chebyshev	1.3 dB	3.0 dB
Bessel	-	-
Inv. Chebyshev	-	-

**Table 8 sensors-23-08057-t008:** Values of the transconductances depending on the selected value of the cut-off frequency.

Theoretical *f*_0_ [kHz]	23.5	47.0	94.0
*g_m_*_1_ [µs]	386.1	784.4	1544.3
*g_m_*_2_ [µs]	136.5	277.3	546.0
*g_m_*_3_ [µs]	93.7	190.3	374.7
*g_m_*_4_ [µs]	96.5	196.1	386.1

**Table 9 sensors-23-08057-t009:** Theoretical, simulated and measured cut-off frequencies based on selected values of transconductances.

Theoretical *f*_0_ [kHz]	23.5	47.0	94.0
Simulated *f*_0_ [kHz]	22.3	46.6	92.9
Measured *f*_0_ [kHz]	20.9	45.2	94.9

**Table 10 sensors-23-08057-t010:** The overshot settling time of the unity step response from Figure 12.

**Approximation**	**Simulations**
Elliptic	317 µs
Butterworth	83 µs
Chebyshev	257 µs
Bessel	59 µs
Inv. Chebyshev	77 µs

**Table 11 sensors-23-08057-t011:** Group delay (Peak Value) of the group delay response from Figure 13.

Approximation	Simulations	Measurement
Elliptic	3.3 µs	2.6 µs
Butterworth	1.3 µs	1.2 µs
Chebyshev	3.5 µs	3.0 µs
Bessel	0.7 µs	0.8 µs
Inv. Chebyshev	1.4 µs	1.3 µs

## Data Availability

Data sharing is not applicable to this article.

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
