# Peer review of "Various-Order Low-Pass Filter with the Electronic Change of Its Approximation"

_sensors, 2023, doi:10.3390/s23198057_

Round 1
Reviewer 1 Report
In this paper, the authors present a low-pass filter structure that allows electronic adjustments of the order (from one to four), cut-off frequency, and approximation type (Bessel, Butterworth, Cauer, Chebyshev, and Inverse Chebyshev). The circuit was implemented using commercially available circuits (EL2082, EL4083, OPA860, and LT1228). To verify the design, the circuit underwent PSpice simulations and experimental measurements.
The Introduction provides an overview of the current state of the art in this field, citing several recent articles on similar topics. However, expanding upon this section could potentially enhance our understanding of the practical applications of these types of circuits.
There is an incorrect statement in line 128 that refers to reference 12. The statement implies the use of "mechanical switches" to change circuit characteristics, but the entire circuit was implemented in a CMOS process. Could you check again?
In the second part of the paper, theoretical information is presented on the proposed reconfigurable/tunable filter, including building blocks, transfer function, and commercially available circuits required for practical implementation. The content is well-documented and presented.
The upcoming sections will concentrate on the design and verification of the proposed topology. There are a few observations about these sections.
The filter was designed for an angular frequency of 300000 rad/s. Is there a specific reason for this choice, or is it just for comparison between PSpice simulation and practical measurements?
In the abstract, PSpice simulations and experimental measurements are mentioned, but only the cutoff frequency data from Table 5 is available. Are the waveforms in Figures 5-11 for qualitative comparison of the circuit's operation? A table with values of the design parameters like operational angular frequency, ripple in the pass-band, stop-band frequency, and attenuation in the stop-band would be helpful.
In section 4.2, only PSpice simulations are presented because obtaining current responses without proper equipment would be difficult. Can you explain further? In section 5, you used a V/I I/V converter for the waveforms. Could a similar approach be used here?
In section 4.4, there is a statement regarding the current gains where all current gains (B) are set to one in line 381. Then, you mention the cut-off frequency, which is for the low-pass filter where only B1 is not null. Can you explain this?
Section 5 tested the filter's response to a sinusoidal signal and a ramp signal, both having a frequency of 20kHz. Theoretically, the sine wave's amplitude should only be affected if the frequency is close to the cut-off frequency, while the ramp signal's shape is altered by any filter provided it has multiple spectral components near the cut-off frequency. The filter was tested at this frequency instead of a lower one with more spectral components in the pass band, similar to the step response, where the frequency used was 1kHz. Could you explain why?
The paper's language is mostly clear, but it contains typos and ambiguous expressions (e.g. "a transfer of the filter" at line 362) that require correction. A comprehensive proofreading should effectively resolve these issues.
Reviewer 2 Report
The paper deals with the reconfigurable various types of LPF approximations. The concept of the paper sounds good, but some points in the paper need to be clarified.
- The scientific contribution of the paper is unclear. Why is the reconfigurable filter essential, and what is the proper application of each approximation?
- The approximations used in the paper are acceptable, but the Cauer is a type of network connection, not the approximation. The approximation types should include Butterworth, Bessel, Chebyshev, and Elliptic approximations.
- The coefficients in Table 2 and Eqs. (3)-(6) are questionable, and it is unclear how to derive and involve the Eqs. (1) and (2)? What are B and b?
- The coefficients of Chebyshev and Cauer approximations (Table 2, 3) are similar, which is confirmed by the results in Fig.5 and 7. As the comment #2, the Cauer should be replaced by Elliptic approximations.
- The experimental results in Figs.19-23 have no merit in clarifying the superior performance of various approximations. The results should be agreed to their performances. For example, Chebyshev has a higher sharpened filtering than Butterworth but is given a higher group delay ripple.
N/A
Round 2
Reviewer 1 Report
Thank you for improving the scientific value of the paper.
Reviewer 2 Report
Thanks for the excellent revision. This version is accepted and suitable for publication.
